# Cellular stress alters 3′UTR landscape through alternative polyadenylation and isoform-specific degradation

Dinghai Zheng [1,2], Ruijia Wang[1,2], Qingbao Ding[1,2], Tianying Wang[3], Bingning Xie[1,2], Lu Wei[1,2], Zhaohua Zhong[3] & Bin Tian [1,2]

Most eukaryotic genes express alternative polyadenylation (APA) isoforms with different 3′UTR lengths, production of which is influenced by cellular conditions. Here, we show that arsenic stress elicits global shortening of 3′UTRs through preferential usage of proximal polyadenylation sites during stress and enhanced degradation of long 3′UTR isoforms during recovery. We demonstrate that RNA-binding protein TIA1 preferentially interacts with alternative 3′UTR sequences through U-rich motifs, correlating with stress granule association and mRNA decay of long 3′UTR isoforms. By contrast, genes with shortened 3′UTRs due to stress-induced APA can evade mRNA clearance and maintain transcript abundance post stress. Furthermore, we show that stress causes distinct 3′UTR size changes in proliferating and differentiated cells, highlighting its context-specific impacts on the 3′UTR landscape. Together, our data reveal a global, 3′UTR-based mRNA stability control in stressed cells and indicate that APA can function as an adaptive mechanism to preserve mRNAs in response to stress.

[1] Department of Microbiology, Biochemistry and Molecular Genetics, Rutgers New Jersey Medical School, Newark, NJ 07103, USA. [2] Rutgers Cancer Institute of New Jersey, Newark, NJ 07103, USA. [3] Department of Microbiology, Harbin Medical University, Harbin, 150081, China. These authors contributed equally: Dinghai Zheng, Ruijia Wang. Correspondence and requests for materials should be addressed to B.T. (email: btian@rutgers.edu)

Eukaryotic cells experience various types of stress under physiological and pathological conditions, such as oxidative stress, heat shock, endoplasmic reticulum stress, etc.[1,2]. Cellular stress has been implicated in aging[3] and various diseases, including cancer[4,5], cardiac conditions[6,7], and neurological disorders[8,9]. As part of adaptive response to stress, certain genes with functions in cell survival and homeostasis are transcriptionally activated under stress[10]. In addition, substantial post-transcriptional regulation takes place in stressed cells. At the center of it is inhibition of translation through phosphorylation of eIF2α[11,12], also known as integrated stress response[13,14]. Inhibited translation is believed to promote formation of stress granules (SGs), which are cytoplasmic, membrane-less structures composed of RNAs and proteins[15,16]. Several RNA-binding proteins (RBPs) play key roles in recruitment of RNAs into SGs, including RasGAP SH3-domain-Binding Protein 1 (G3BP1) and T cell-restricted intracellular antigen-1 (TIA1)[16]. While SGs are generally considered to be beneficial to cell survival[17], they can also lead to more permanent granule structures in the cell with adverse outcomes, which has been implicated in neurodegenerative diseases[16].

Over 70% of mammalian mRNA genes harbor multiple polyadenylation sites (PASs), resulting in expression of alternative polyadenylation (APA) isoforms with different 3′UTR lengths[18–20]. Short and long 3′UTRs of a given gene can differ considerably in length[21]. As such, alternative 3′UTR isoforms can have quite different mRNA metabolisms, such as subcellular localization, mRNA stability, and translation[19,20]. APA isoform profiles often display global, directional differences between cell types and conditions. For example, transcripts expressed in the nervous system tend to have longer 3′UTRs than in other tissue types, whereas short 3′UTRs are highly abundant in testis[22–24]. In addition, proliferating cells tend to use proximal PASs, as compared to quiescent and differentiated cells[25–27], and 3′UTRs generally lengthen during embryonic development[26]. Moreover, various cellular conditions, such as cell senescence[28], neuronal activation[29–31], and viral infections[32,33], have been shown to alter global APA profiles in the cell.

Different types of stress have been shown to impact APA in yeast[34,35], plants[36], and mammalian cells[37–39]. However, the consequences of APA on mRNA metabolism in stressed cells are unclear, and how APA changes during recovery from stress is completely unknown. Here, we examine APA profiles of steady state and newly made RNAs in cells under stress and during recovery from stress. We compare mRNA stability and interactions with the RBP TIA1 between 3′UTR isoforms under normal and stress conditions. We provide evidence showing the importance of 3′UTR length and sequence motifs for gene expression in stressed cells and indicate that stress-induced APA is an adaptive mechanism that preserves mRNAs in response to stress.

## Results

**Arsenic stress elicits global 3′UTR shortening**. We were interested in APA profiles in cells under stress and during recovery from stress. We treated mouse NIH3T3 cells with 250 μM sodium arsenite (NaAsO$_2$), a commonly used strong stressor, for 1 h, and let cells recover for 4, 8, 12, or 24 h after removal of arsenic stress (AS) (Fig. 1a). As expected, while cell death was not detected, cells after stress displayed significantly slowed growth (Supplementary Fig. 1). In line with the integrated stress response mechanism[14], the level of eIF2α phosphorylation increased by ~7-fold after 1 h of AS and went back to the baseline level after 4 h of recovery (RC, Fig. 1b), indicating successful stress response and recovery. Consistently, SGs were detected by immunocytochemistry using

anti-PABPC1 antibody after 1 h of AS, and disappeared after 4 h of recovery (Fig. 1c).

To examine APA, we extracted total cellular RNA from non-treated (NT) control cells, cells with 1 h of AS, and those at different time points of recovery (4–24 h), and subjected them to 3′ region extraction and deep sequencing (3′READS), a method specialized for interrogation of 3′ ends of poly(A)+ transcripts[21,40]. To simplify APA analysis, we focused on the two most abundant 3′UTR APA isoforms as determined by PAS reads (see Methods for details). Based on their relative locations in the 3′UTR, they were named proximal PAS (pPAS) and distal PAS (dPAS) isoforms, respectively (illustrated in Fig. 1d, top). We found that the number of genes showing shortened 3′UTRs in AS-treated cells (pPAS isoform abundance relative to that of dPAS isoform increased) was 2.3-fold greater than the number of genes showing the opposite trend (247 vs. 108, Fig. 1d, bottom), indicating that AS leads to general 3′UTR shortening in the cell. The global 3′UTR APA changes were also measured by the relative expression difference (RED) score, which was the difference between two samples in ratio (log2) of dPAS isoform abundance (reads per million, RPM) to pPAS isoform abundance. A negative RED value indicates 3′UTR shortening and a positive value 3′UTR lengthening. To robustly compare samples with different sequencing depths, we used the Global Analysis of Alternative Polyadenylation (GAAP) method we previously developed[41], which employs random sampling of reads from comparing samples to derive a median RED and its standard deviation (see Methods for details). Using GAAP, the median RED of cells with AS was −0.11 (Fig. 1e, red bar).

Interestingly, 3′UTR shortening became more conspicuous after 4 h of recovery (median RED = −0.27, $P = 1.5 \times 10^{-11}$ vs. 1 h AS, Wilcoxon test, Fig. 1e). The median RED score gradually increased afterwards (median RED = −0.14 and −0.12 for 8 and 12 h of recoveries, respectively, Fig. 1e) and was similar to that of NT cells after 24 h (median RED = 0.03, Fig. 1e).

We found that the extent of 3′UTR shortening of a gene, as measured by the RED score, was a function of the distance between pPAS and dPAS, or size of alternative UTR (aUTR, illustrated in Fig. 1d, top), in both AS and RC cells (Fig. 1f). That is the longer the aUTR, the greater the 3′UTR shortening. The gene group with the longest aUTRs (bin 5 in Fig. 1f) displayed significantly greater shortening than the group with the shortest aUTRs (bin 1 in Fig. 1f) in both AS and RC cells ($P = 2.7 \times 10^{-20}$ and $5.1 \times 10^{-29}$, respectively, Wilcoxon test).

An example gene Nmt1 (encoding N-myristoyltransferase 1) is shown in Fig. 1g, which expressed two 3′UTR APA isoforms with 3′UTR lengths of 337 nt and 2.8 kb, respectively. Using reverse transcription-quantitative PCR (RT-qPCR) and primer pairs targeting a region common to both pPAS and dPAS isoforms as well as the aUTR, we validated 3′UTR shortening of Nmt1 in cells with AS and in recovery, as well as those of Calm1, Purb, and Timp2 (Fig. 1h). Notably, 3′UTR shortening could also be detected in cells treated with oxidative stress inducer H$_2$O$_2$ (Supplementary Figs. 2a and 2b) or with a lower level of AS (25 μM) (Supplementary Figs. 2c and 2d), although the timing and the degree of shortening were different.

**Analysis of newly made RNAs reveals stress-induced APA**. Shortening of 3′UTRs can be caused by increased usage of the PASs of short 3′UTR isoforms or by enhanced degradation of long 3′UTR isoforms, or both. We reasoned that analysis of APA profiles with newly made RNAs would reveal PAS usage changes[42]. To this end, we added 4-thiouridine (4sU) to cell culture media 1 h before harvesting cells in NT, AS (1 h), or RC (4 h) conditions (Fig. 2a), separated 4sU-labeled RNAs from

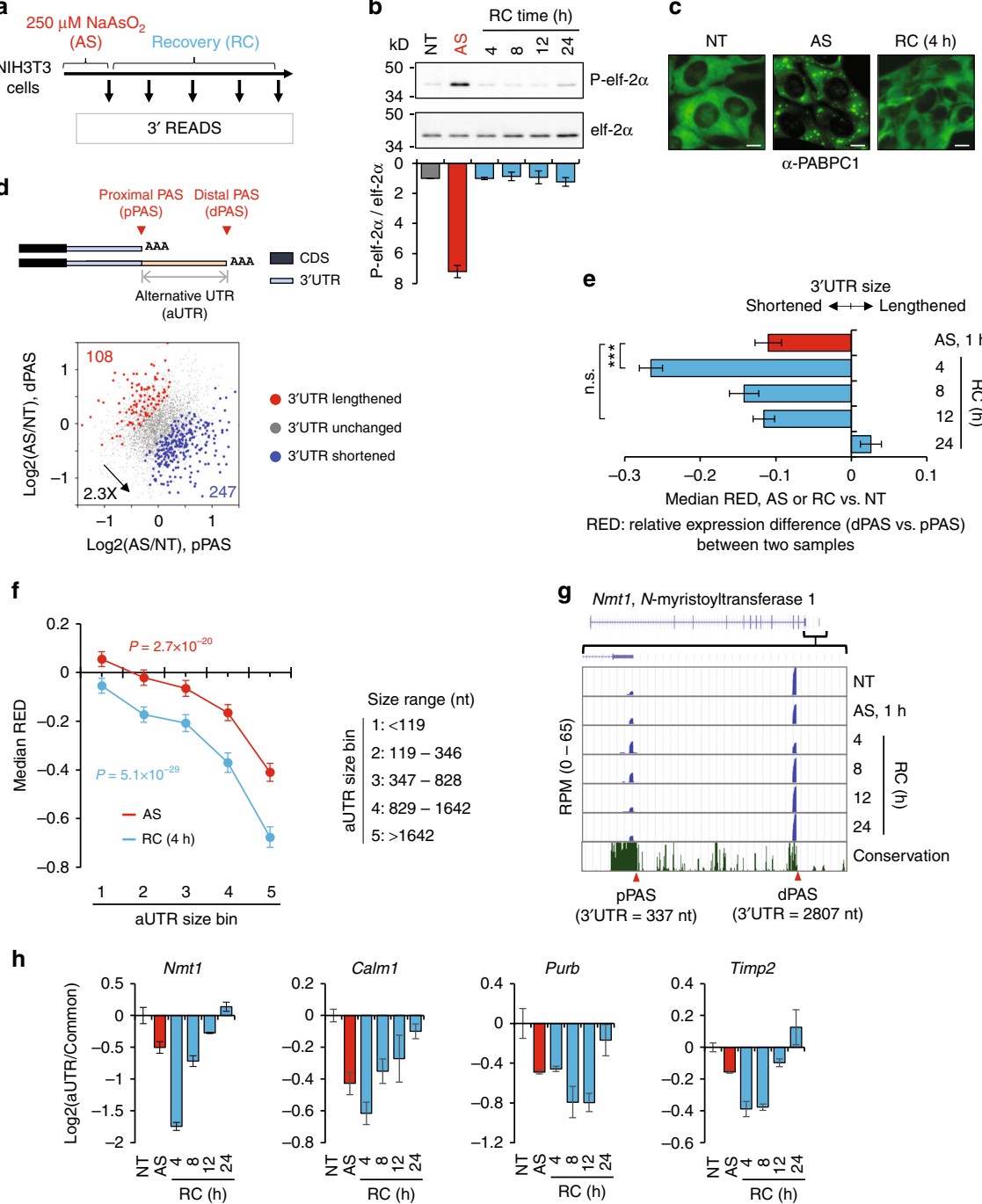

**Fig. 1** Arsenic stress elicits global 3'UTR shortening. **a** Schematic of experimental design. AS, 1 h treatment with 250 μM sodium arsenite; RC, recovery after AS. **b** Top, western blot analysis of phosphorylated (upper) and total (lower) eIF-2α protein. NT non-treated. Bottom, normalized ratio of amount of phosphorylated eIF-2α to total eIF-2α. Error bars are standard deviation based on two replicates. **c** Immunocytochemistry analysis of stress granules (SGs) using anti-PABPC1. Scale bar: 10 μm. **d** Top, schematic of two APA isoforms using proximal PAS (pPAS) or distal PAS (dPAS) in the 3'UTR. The region between the two PASs is named alternative 3'UTR (aUTR). Bottom, scatter plot showing expression change of pPAS isoform (x-axis) and that of dPAS isoform (y-axis) in total RNA after 1 h of AS. Genes with significantly shortened or lengthened 3'UTRs (P < 0.05, Fisher's exact test) based on two biological replicates are highlighted in blue and red, respectively. **e** Median relative expression difference (RED) values of treatment samples compared to the NT sample, which reflect the degree of global 3'UTR length changes. Significance of difference (Wilcoxon test) is indicated, with three asterisks indicating P < 0.001, and n.s. P > 0.05. Error bars are standard deviations based on random sampling of data for 20 times. **f** Relationship between RED and aUTR size. Genes were divided into five bins based on aUTR size. Median RED for each gene bin is shown for AS (red line) or RC (blue line) vs. NT cells. Error bars are standard error of mean. P-values (Wilcoxon test) are based on difference between gene bins 1 and 5. **g** 3'READS data of an example gene Nmt1, visualized in UCSC genome browser. Expression levels of isoforms are indicated by reads per million (RPM) PAS reads. Note the RefSeq track at the top does not cover the dPAS due to poor annotation. **h** RT-qPCR analysis of the relative amounts of APA isoforms (x-axis). Two primer sets were used for each gene, targeting a common region to both isoforms and the aUTR, respectively. The log2(aUTR/common) value (y-axis) indicates relative abundance of long vs. short 3'UTR isoforms. Error bars are standard deviation based on two replicates

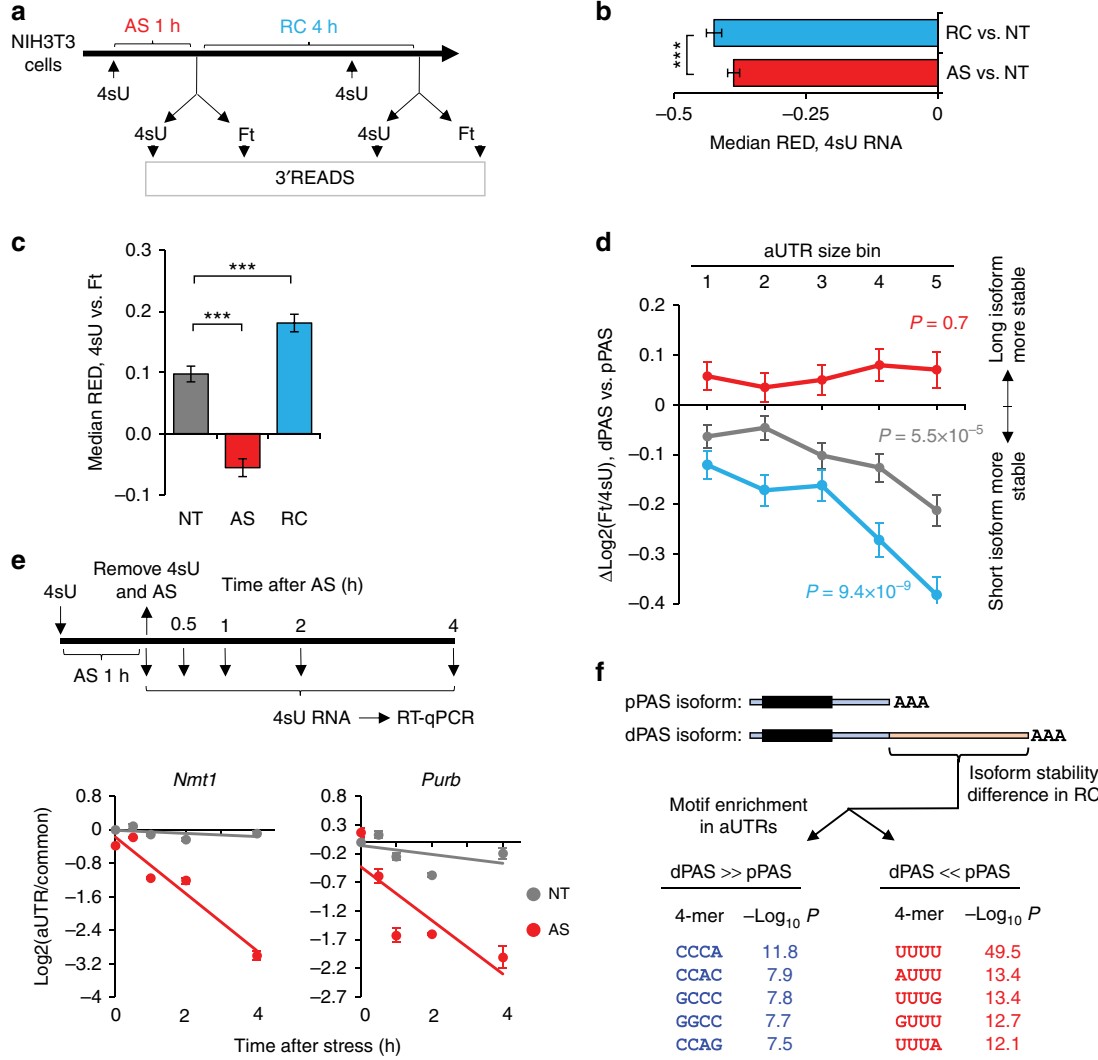

**Fig. 2** Both preferential expression of short 3′UTR isoforms and degradation of long 3′UTR isoforms take place in stressed cells. **a** Schematic of experimental design. Cells were labeled with 4sU for 1 h right before harvest during arsenic stress (AS) or recovery (RC), as well as in non-treated control (NT) cells. 4sU-labeled and flow-through (Ft) RNAs were subject to APA analysis with 3′READS. **b** Comparison of global 3′UTR length changes between AS or RC cells and NT cells using RNA in the 4sU fraction only. The NT result was based on two replicates, and AS and RC results on one sample each. Error bars are standard deviation of median RED based on random sampling of the data for 20 times (see Methods for details). Three asterisks indicate $P < 0.001$ (Wilcoxon test). **c** Comparison of global 3′UTR length differences between the 4sU and Ft fractions, as indicated by median RED, under NT, AS (1 h), and RC (4 h) conditions. The NT result was based on two replicates, and AS and RC results on one sample each. Error bars are standard deviation of median RED as in **b**. Three asterisks indicates $P < 0.001$ (Wilcoxon test). **d** Stability difference between 3′UTR APA isoforms, as measured by $\Delta\log2($Ft/4sU$)$ between dPAS and pPAS isoforms, for gene groups with different aUTR sizes. Genes were divided into five bins as in Fig. 1f. The median $\Delta\log2($Ft/4sU$)$ value for each gene bin is plotted for NT (gray), AS (red), or RC (blue) conditions. Error bars are standard error of mean based on all genes in each bin. $P$-values (Wilcoxon test) comparing genes in bin 1 vs. in bin 5 are shown. **e** Validation of mRNA decay analysis. Top, schematic of experimental design. NIH3T3 cells were subject to 1 h of AS, during which 4sU was used to label newly made RNA. 4sU-labeled RNA was analyzed by RT-qPCR at different time points after removal of stress and 4sU. Bottom, RT-qPCR result with primer sets targeting aUTR and common regions of APA isoforms. NT cells were used as a control. **f** Sequence motif analysis of aUTR sequences of genes whose long 3′UTR isoforms were more stable than short 3′UTR isoforms (dPAS » pPAS, genes with top 10% of $\Delta\log2($Ft/4sU$)$, dPAS vs. pPAS in RC) or the opposite (dPAS « pPAS, genes with bottom 10% of $\Delta\log2($Ft/4sU$)$, dPAS vs. pPAS in RC) in the RC sample. Top five enriched tetramers in each group of genes are shown. Their $P$-values (Fisher's exact test) are indicated

non-labeled, or flow-through (Ft) RNAs and subjected both RNA pools to 3′READS analysis (Supplementary Fig. 3a, see Methods for details). As such, newly made RNAs and pre-existing RNAs were enriched in the 4sU and Ft fractions, respectively.

Similar to the result with steady-state RNAs, 4sU-labeled RNAs displayed 3′UTR shortening in both RC and AS cells as compared with NT cells (Fig. 2b). However, the difference in 3′UTR shortening between the two conditions, although still significant, was much smaller in newly made RNAs ($\Delta$Median RED = 0.03;

$P = 4.4 \times 10^{-10}$, Wilcoxon test, Fig. 2b), as compared to the difference using steady state RNAs ($\Delta$Median RED = 0.16, Fig. 1e).

We next compared RED scores between 4sU and Ft fractions. In control cells, 3′UTRs were longer in newly made RNAs than in pre-existing RNAs (median RED = 0.10, Fig. 2c), consistent with the notion that short 3′UTR isoforms are generally more stable than long 3′UTR isoforms[43]. A similar trend was observed with RC cells, but to a greater degree (median RED = 0.18, Fig. 2c). By

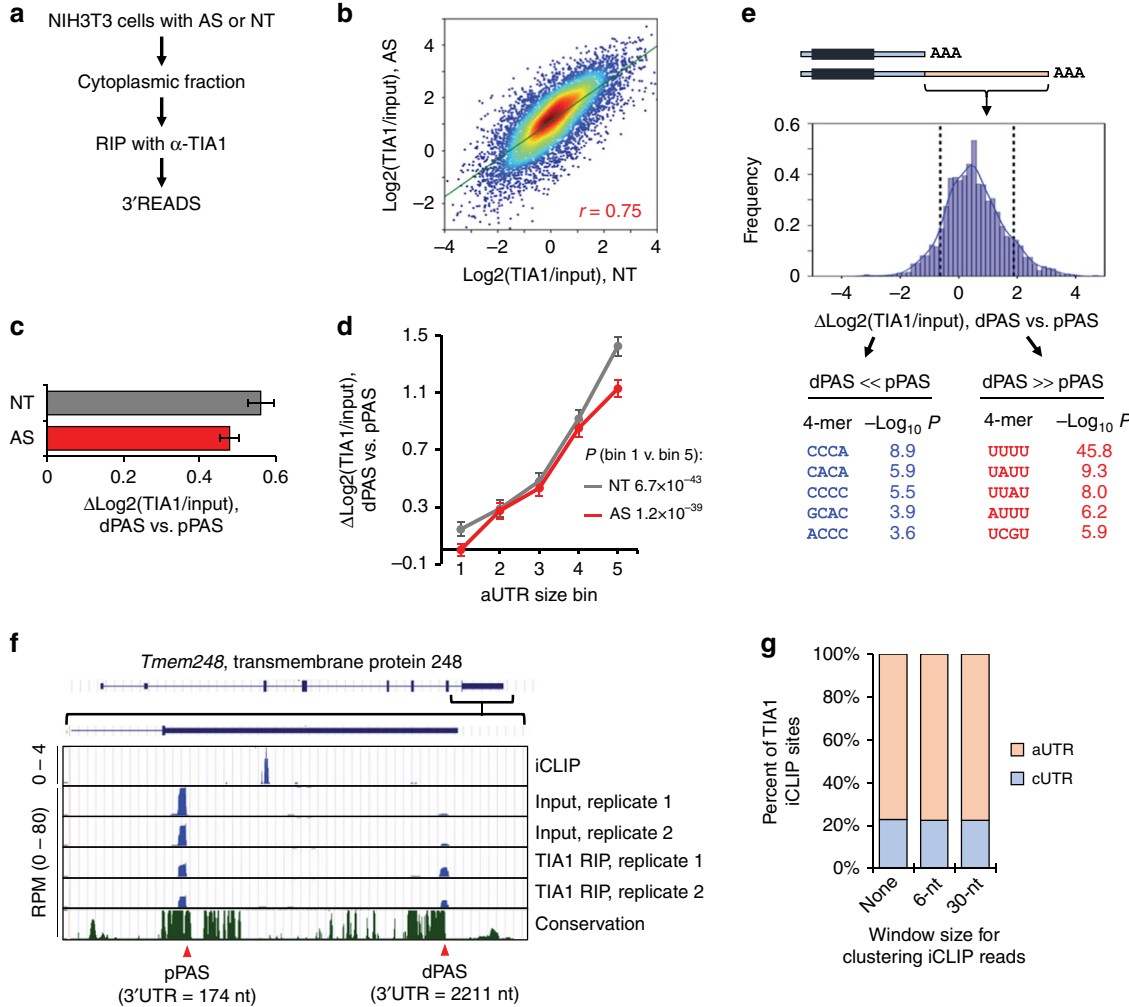

**Fig. 3** TIA1 preferentially binds to long aUTRs via U-rich motifs. **a** Schematic of 3′READS + RIP. The cytoplasmic fraction of NT or AS cells was used for ribonucleoprotein immunoprecipitation (RIP) with a TIA1 antibody. Two replicates were used for both input and RIP samples. **b** Correlation of log2(TIA1/input) values in NT and AS cells. Pearson correlation coefficient $r$ is indicated. **c** Differences in TIA1 binding between distal pA isoforms (dPAS) and proximal pA isoforms (pPAS), as measured by median $\Delta$log2(TIA1/Input), dPAS vs. pPAS, under NT (gray) or AS (red) condition. Error bars are standard deviation of median RED based on random sampling of the data for 20 times (see Methods for details). **d** Relationship between $\Delta$log2(TIA1/Input) and aUTR size. Genes were divided into five bins based on aUTR size as in Fig. 1f. Error bars are standard error of mean based on all genes in each bin. $P$-values (Wilcoxon test) for comparing genes in bin 1 vs. in bin 5 are shown. **e** Top, distribution of $\Delta$log2(TIA1/Input) values representing TIA1 binding difference between long and short 3′UTR isoforms under AS. Top and bottom 10% genes were selected for sequence motif analysis. Bottom, tetramers enriched in the aUTRs of top 10% of genes (dPAS » pPAS), or of bottom 10% of genes (dPAS « pPAS). Top five enriched tetramers are shown for each group of genes. Their $P$-values (Fisher's exact test) are indicated. Only the data from AS cells were used. **f** TIA1 iCLIP and RIP-3′READS data of an example gene *Tmem248*, as shown in UCSC genome browser tracks. The aUTR of *Tmem248* contains TIA1 binding sites as indicated in the iCLIP track. *Tmem248* has two 3′UTR APA sites, as indicated in the input and RIP tracks, resulting in two 3′UTR isoforms with a 174- or 2211-nt 3′UTR. Transcript expression levels are indicated by reads per million (RPM) PAS reads. The dPAS isoform is more enriched in the RIP fraction than in the cytoplasmic input, as compared to pPAS isoform. **g** Percentages of 3′UTR TIA1 iCLIP sites in cUTRs (blue) and aUTRs (pink) using different window sizes for clustering iCLIP reads

contrast, newly made RNAs in AS cells had shorter 3′UTRs than pre-existing RNAs (median RED = −0.06, Fig. 2c). These results indicate that 3′UTR shortening in cells with 1 h AS treatment is largely due to newly made RNAs, supporting the notion that AS leads to preferential usage of proximal PASs. While this also takes place in RC cells (Fig. 2b), the much greater shortening of 3′UTRs in those cells appears to be attributable to the pre-existing pool of RNA (Fig. 2c), suggesting differential regulation of mRNA stability between 3′UTR isoforms.

**Enhanced 3′UTR size-based mRNA decay after stress.** To specifically address how mRNA stability contributed to 3′UTR APA profiles, we calculated for each isoform the ratio of its abundance

in the Ft fraction to that in the 4sU fraction, or log2(Ft/4sU). After adjustments for the number of uridines in each transcript to remove its influence on the abundance of 4sU-labeled RNAs (Supplementary Fig. 3b), the log2(Ft/4sU) values in control cells correlated well with mRNA half-lives previously measured in NIH3T3 cells by Spies et al.[43] ($r = 0.52$, Pearson correlation, Supplementary Fig. 3c), supporting the suitability of using log2(Ft/4sU) values to reflect mRNA stability.

To examine the influence of 3′UTR size on stability, we divided genes into five bins based on their aUTR sizes. We found that long 3′UTR isoforms in control cells were generally less stable than short 3′UTR isoforms, as indicated by the negative median $\Delta$log2(Ft/4sU) values in all gene groups (Fig. 2d, gray line). The difference in stability between short and long 3′UTR isoforms was

a function of the aUTR size ($P = 5.5 \times 10^{-5}$ comparing genes in bins 1 and 5, Wilcoxon test, Fig. 2d, gray line), indicating a negative role of 3′UTR size in mRNA stability. Significantly, the difference between long and short 3′UTR isoforms became greater in cells during recovery (blue vs. gray lines, Fig. 2d), indicating enhanced 3′UTR size-based mRNA decay. Consistently, the difference in stability between gene bins 1 and 5 was much greater in RC cells ($P = 9.4 \times 10^{-9}$, Wilcoxon test, Fig. 2d, blue line). By contrast, in AS cells $\Delta$log2(Ft/4sU) values were positive across all bins and were not correlated with aUTR sizes (red line in Fig. 2d), indicating general stabilization of distal PAS isoforms during stress. The stability difference between short and long 3′UTR isoforms in control cells and in cells after stress was validated by RT-qPCR analysis of mRNA decays of APA isoforms of *Nmt1* and *Purb* (Fig. 2e).

We next asked whether some sequence motifs in 3′UTRs were associated with mRNA decay during recovery from stress. To this end, we identified two groups of genes (Fig. 2f, top), i.e., genes whose long 3′UTR isoforms were less stable than short 3′UTR isoforms (group 1) and those showing the opposite trend (group 2). We then compared tetramer frequencies in the aUTRs of these two gene groups. Interestingly, U-rich tetramers were significantly enriched in the aUTRs of group 1 genes, whereas C-rich elements in those of group 2 genes (Fig. 2f, bottom), indicating involvement of sequence motifs in mRNA decay in cells after stress.

Together, our data indicate that long 3′UTR isoforms are generally less stable than short 3′UTR isoforms in control cells; this trend is blunted in cells under stress but is accentuated during recovery from stress. In addition, U-rich motifs play a prominent role in mRNA decay during recovery from stress.

**Long isoforms bind TIA1 more efficiently** via **U-rich motifs**. Sequence motif analysis of unstable mRNAs suggested that U-rich motif-binding RBPs might be responsible for the enhanced 3′UTR-based mRNA decay in cells after stress. We next focused on TIA1 because of its activity in binding U-rich motifs and role in recruiting mRNAs into SGs during stress[16]. As reported, TIA1 was mostly in nucleus in control cells and became enriched in SGs during stress (Supplementary Fig. 4a). Four hours of recovery after AS restored the TIA1 subcellular distribution (Supplementary Figure 4a). Notably, TIA1 protein levels were unchanged across these conditions (Supplementary Fig. 4b).

To examine interactions between mRNAs and TIA1, we carried out ribonucleoprotein immunoprecipitation (RIP) using an anti-TIA1 antibody (Supplementary Fig. 4b) to isolate TIA1-bound mRNAs from cytoplasm of AS-treated or control cells, followed by 3′READS (illustrated in Fig. 3a). As such, this approach, named 3′READS + RIP, could distinguish APA isoforms in TIA1 binding.

Using input cytoplasmic RNA as a reference, we calculated TIA1 binding efficiency, as calculated by log2(TIA1/Input), for each transcript with a defined PAS (see Methods for details). Two biological replicates were generated, which were well correlated ($r = 0.65$ and 0.72 for NT and AS samples, respectively, Pearson correlation, Supplementary Fig. 4c). In addition, log2(TIA1/Input) values were well correlated between AS-treated and control cells ($r = 0.75$, Pearson correlation, Fig. 3b), indicating that TIA1 binding efficiency with its target RNAs generally does not change by AS. Also notable is that the TIA1 binding efficiency measured by 3′READS + RIP is generally correlated with the number of TIA1 binding sites in 3′UTRs as measured by a recent iCLIP study using B cells[44] ($r = 0.34$, Supplementary Fig. 4d), indicating similarities in TIA1 binding across cell types and supporting the suitability of using 3′READS + RIP for quantitative analysis of TIA1 binding.

Interestingly, we found that TIA1-bound transcripts tended to have significantly longer 3′UTRs in both control and AS cells, as indicated by the positive median $\Delta$log2(TIA1/Input) values measuring the difference between dPAS and pPAS isoforms (Fig. 3c) and that genes whose long 3′UTR isoform had a higher log2(TIA1/Input) value than short isoform markedly outnumbered genes with the opposite trend (4.7- and 5.1-fold for NT and AS cells, Supplementary Fig. 4e). Consistently, the $\Delta$log2(TIA1/Input) value between long and short 3′UTR isoforms was a function of the aUTR size in both NT and AS cells (Fig. 3d). In line with these results, the log2(TIA1/Input) values of all transcripts correlated with their 3′UTR size ($r = 0.62$, Pearson correlation, Supplementary Fig. 4f), and 3′UTR size was a better determinant than other transcript features, such as 5′UTR and CDS sizes, for interaction with TIA1 based on a linear regression model (Supplementary Fig. 4g). These results indicate that the efficiency of TIA1 binding to an mRNA positively correlates with its 3′UTR size.

We next examined sequence motifs in aUTRs of two gene groups, i.e., genes whose long 3′UTR isoforms had a higher TIA1 binding efficiency than short 3′UTR isoforms (group 1, dPAS » pPAS in Fig. 3e), and those with the opposite trend (group 2, dPAS « pPAS in Fig. 3e). Consistent with the known TIA1 binding properties[44], U-rich motifs, especially UUUU, were significantly enriched in the aUTRs of group 1 genes (Fig. 3e, bottom). An example gene *Tmem248* is shown in Fig. 3f, which expressed a 174-nt, short 3′UTR isoform and a 2,211-nt, long 3′UTR isoform. A prominent TIA1 binding site was located in its aUTR, as identified by the iCLIP study[44]. Consistently, the long 3′UTR isoform was markedly enriched by 3′READS + RIP as compared to the short 3′UTR isoform (Fig. 3f). In addition, using the iCLIP data and all mouse PASs in the PolyA_DB database[45], we found that about 80% of TIA1 binding sites in 3′UTRs were located in aUTRs (Fig. 3g). Taken together, these results indicate that long 3′UTR isoforms preferentially bind TIA1 as compared to short 3′UTR isoforms through U-rich motifs in alternative 3′UTR sequences.

**TIA1 binding correlates with mRNA decay and SG association**. We found that TIA1 binding efficiency of a transcript, as measured by log2(TIA1/Input), was negatively correlated with its mRNA stability as measured by log2(Ft/4sU)($r = -0.56$, Pearson correlation, Fig. 4a), suggesting a connection between TIA1 binding and mRNA decay. Importantly, the transcripts with both strong TIA1 binding and a short half-life tended to have significantly longer 3′UTRs than those with both weak TIA1 binding and a long half-life (Fig. 4b). In addition, transcripts with strong TIA1 binding in AS cells tended to have significantly decreased mRNA stability during recovery ($\Delta$log2(Ft/4sU), RC vs. AS) than transcripts with weak TIA1 binding ($P = 6.0 \times 10^{-25}$, K–S test, Fig. 4c). More importantly, long 3′UTR isoforms which had higher TIA1 binding efficacies in AS cells than short 3′UTR isoforms were significantly less stable in RC cells than the latter (Fig. 4d). Together, these data indicate that TIA1 binding correlates with enhanced mRNA decay during recovery from stress.

A recent study identified mRNAs in the SG core through isolation of RNAs bound to G3BP1, another resident RBP in SGs[46]. We thus asked whether TIA1 binding efficiency was related to enrichment in the SG core. To this end, we obtained the SG-enrichment data of human transcripts expressed in U-2 OS cells that were generated by Khong et al.[46] and compared them to TIA1 binding efficiencies of their mouse orthologs calculated in this study (see Methods for details). As shown in Fig. 4e, SG-enrichment was generally correlated with TIA binding in AS cells ($r = 0.58$, Pearson correlation), supporting the role of TIA1 in

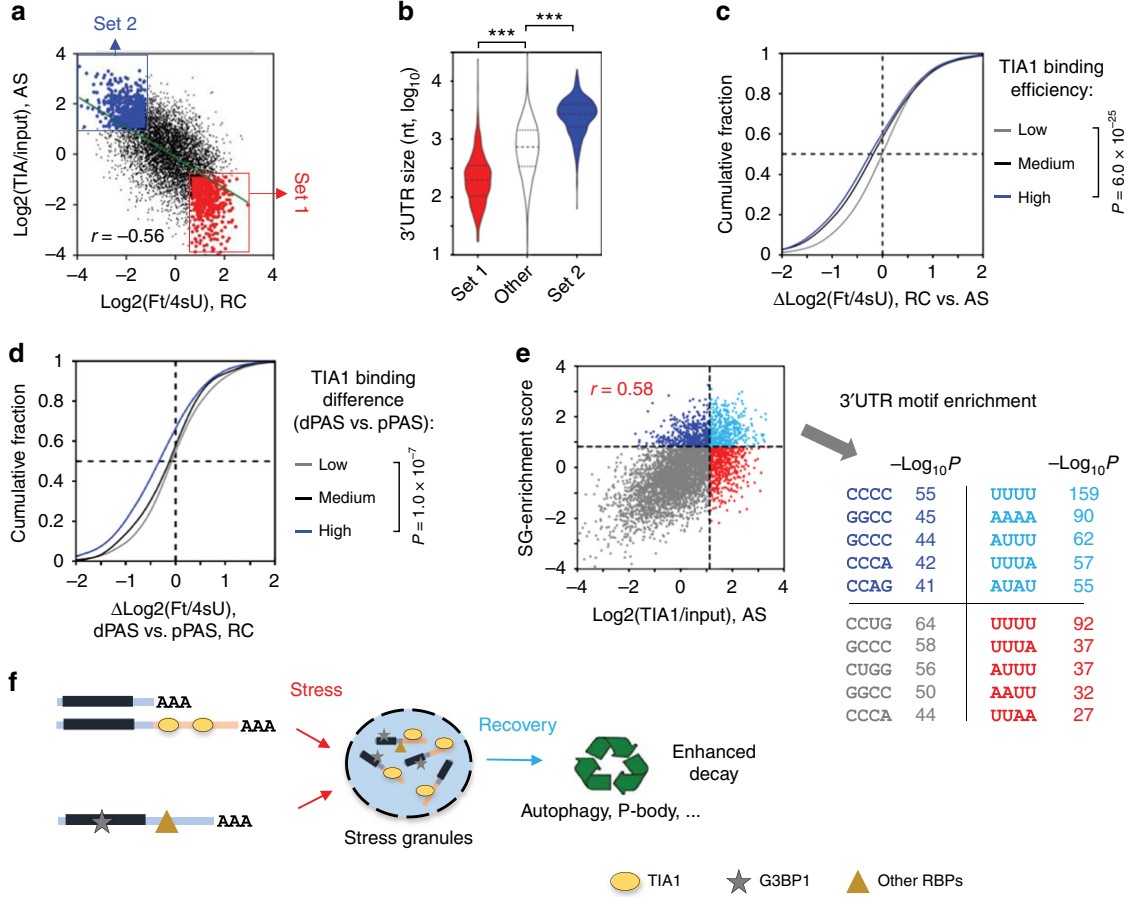

**Fig. 4** TIA1 binding correlates with SG enrichment and mRNA decay after stress. **a** Comparison of TIA1 binding in AS cells and mRNA stability in RC cells. Each dot is a transcript with a defined PAS. Red dots (gene set 1) are transcripts with low TIA1 binding in AS (bottom 20%) and high stability in RC (top 20%). Blue dots (gene set 2) are transcripts with high TIA1 binding in AS (top 20%) and low stability in RC (bottom 20%). **b** Violin plots of 3′UTR length for three transcript sets defined in **a**. Significance of difference between transcript sets is indicated (***, $P < 0.001$, Wilcoxon test). **c** Cumulative distribution of difference in mRNA stability between RC and AS cells ($\Delta$log2(Ft/4sU)) for transcripts with different TIA1 binding efficiencies. High, medium, and low TIA1 binding transcripts were top 1/3, middle 1/3, and bottom 1/3 based on log2(TIA1/Input), respectively. **d** Cumulative distribution of difference in stability between APA isoforms in RC ($\Delta$Log2(Ft/4sU, RC), dPAS vs. pPAS) for genes with low (gray), medium (black), or high (blue) difference in TIA1-binding between APA isoforms ($\Delta$Log2(TIA1/Input, AS), dPAS vs. pPAS). **e** Left, scatter plot showing comparison of TIA1 binding and SG enrichment under AS. Each dot represents a mouse gene and its homologous human gene. SG-enrichment score is based on association of transcripts with G3BP1 in the SG core, and TIA1 binding is log2(TIA1/Input) in AS cells. Right, enriched tetramers in the 3′UTRs of genes from different groups (as colored in the scatter plot). **f** A model summarizing TIA1 binding to 3′UTR isoforms, SG association during stress, and mRNA decay during recovery. Some transcripts are recruited to SGs independent of TIA1

recruiting mRNAs to SGs and indicating that the recruitment is conserved between human and mouse orthologs. Importantly, U-rich motifs were significantly enriched in the 3′UTRs of transcripts with both high SG-enrichment scores and high TIA1 binding efficiencies (light blue dots in Fig. 4e). U-rich motifs were also enriched, albeit with a lower significance, for transcripts with high TIA1 binding efficiencies but modest SG-enrichment scores (red dots in Fig. 4e). By contrast, the 3′UTRs of transcripts with high SG-enrichment scores but modest TIA1 binding efficiencies (dark blue dots in Fig. 4e) had no enrichment of U-rich motifs. This result indicates that U-rich motifs facilitate TIA1 binding leading to efficient SG recruitment, and some transcripts can be recruited to SGs through TIA1-independent pathways. Taken together, as summarized in Fig. 4f, our data reveal a TIA1-based SG recruitment mechanism in cells under stress, which leads to enhanced mRNA decay during recovery.

**3′UTR shortening through APA enhances gene expression**. We next reasoned that if short 3′UTR isoforms had a higher mRNA stability than long isoforms in cells after stress, the genes with shortened 3′UTRs in response to stress should have a higher transcript abundance after stress than other genes. To this end, we measured RNA abundances in NT, AS, and RC cells using a strand-specific RNA-seq method (Fig. 5a, see Methods for details). We divided genes into three groups, namely, genes with significantly shortened 3′UTRs through APA by stress (group 1), genes expressing 3′UTR APA isoforms but without significant 3′UTR shortening (group 2), and genes showing only one 3′UTR isoform (no APA, group 3). We defined group 1 genes as those showing 3′UTR shortening in both the 4sU fractions of AS and RC cells (blue dots in Fig. 5b).

All three gene groups had a median log2(ratio of gene expression change) of 0 after 1 h of AS, indicating no global expression changes. By contrast, after 4 h of recovery, group 1 genes showed a global upregulation of expression (median log2(ratio) = 0.26), significantly higher than groups 2 and 3 genes (median log2(ratio) = 0.04 and −0.14, respectively; $P = 3.6 \times 10^{-6}$ and $7.6 \times 10^{-15}$, respectively, K–S test, Fig. 5c).

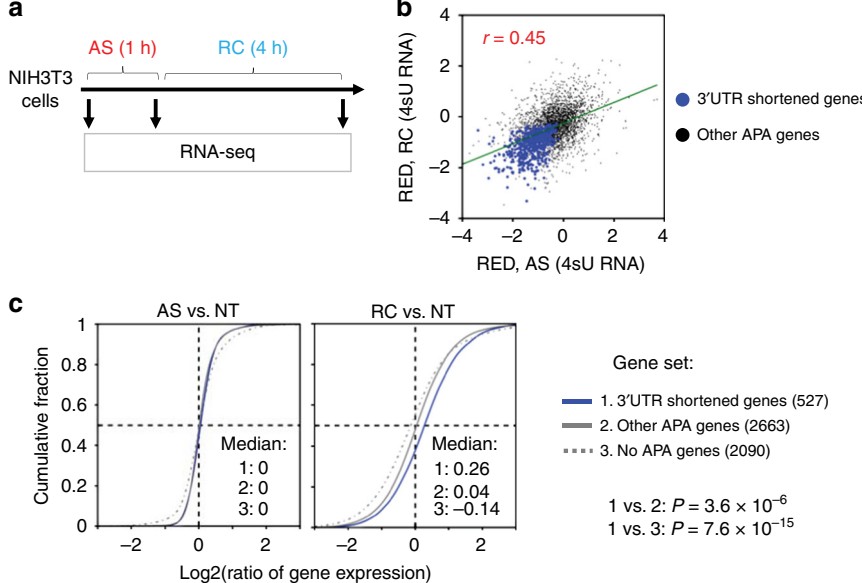

**Fig. 5** 3′UTR shortening by APA leads to preservation of mRNA during stress recovery. **a** Schematic of experimental design. Total cellular RNA was subject to strand-specific RNA-seq to examine gene expression changes. Only reads mapped to CDS were used, which avoided the influence of 3′UTR size change on gene expression analysis. **b** Identification of genes with shortened 3′UTRs through APA. Genes with significant 3′UTR shortening in newly made RNAs of both AS and RC samples (shown in blue) were selected. Black dots are other genes with APA isoforms detected in NIH3T3 cells. **c** RNA abundance changes for three groups of genes after AS (1 h, left) or RC (4 h, right). Groups 1 and 2 genes correspond to blue and black dots in **b**. Group 3 genes were those without APA isoform expression in NIH3T3 cells. Gene numbers are shown in parentheses. Median log2(ratio) of gene expression level (RPM) between two conditions is shown in each graph. *P*-values (K–S test) indicating difference in gene expression change (RC vs. NT) between gene groups are shown

This result indicates that AS-induced 3′UTR shortening can indeed preserve transcripts during recovery from stress. Because group 3 genes, but not group 2 genes, showed general downregulation of gene expression, this result also indicates that having APA sites can help genes maintain expression after stress. Interestingly, Gene Ontology (GO) analysis indicated that genes with functions in cell differentiation, cellular component assembly, signal transduction, and cell proliferation were enriched for group 1 genes (Table 1), indicating that these genes are more likely to use APA to evade mRNA degradation in stressed cells. By contrast, no GO terms were found to be significantly associated with genes showing lengthened 3′UTRs.

To further validate APA changes by stress and its impact on gene expression, we resorted to reporter assays using the pRiG vectors we previously constructed[26]. Due to APA, the pRiG-AD vector could express two isoforms: a long isoform encoding both red fluorescent protein (RFP) and enhanced green fluorescent protein (EGFP), and a short isoform encoding RFP only (Fig. 6a). Twelve hours after transfection, we subjected cells transfected with pRiG-AD to AS for 1 h and measured abundances of two APA isoforms by Northern blot 12 h later (Fig. 6b). Consistent with the global trend of cellular genes, we found that the ratio of short isoform abundance to that of the long isoform increased by 1.4-fold in AS-treated cells (Fig. 6c).

In agreement with the Northern blot data, analysis of cells with fluorescence-activated cell sorting (FACS, Supplementary Fig. 5) showed a 2.8-fold increase of the red to green fluorescence ratio (log2(red/green)) in cells transfected with pRiG-AD and treated with AS, as compared to transfected cells without treatment (Fig. 6d, middle). By contrast, a much milder increase (1.1-fold, Fig. 6d, top) was observed in cells transfected with the pRiG vector, which expressed the long isoform only due to absence of a PAS between the RFP and EGFP sequences[47].

We also transfected cells with a mixture of pRiG and pRiG-SV40 vectors, the latter of which predominantly expresses the short APA isoform owing to the presence of a strong SV40 early PAS between the RFP and EGFP sequences[47]. As such, while both short and long isoforms were produced in cells transfected with pRiG + pRiG-SV40, the isoforms were expressed from two separate plasmids. We adjusted the ratio of the two plasmids to ensure similar amounts of short and long isoforms expressed in the transfected cells to cells transfected with pRiG-AD. The pRiG + pRiG-SV40 mixture yielded an increase of 1.6-fold in log2(red/green) in stressed vs. control cells (Fig. 6d, bottom), substantially lower than that of pRiG-AD. This result indicates that enhanced usage of proximal PAS contributes to the increased log2(red/green) value in pRiG-AD-transfected cells after stress. On the other hand, the difference between cells transfected with pRiG and those with pRiG + pRiG-SV40 indicates that short 3′UTR isoforms can have higher protein expression potentials than long 3′UTR isoforms in cells after stress.

To further validate the effect of 3′UTR APA on protein expression, we cloned full 3′UTR sequences of *Nmt1*, *Timp2*, and *Dnajb1* into the psiCHECK2 vector (Fig. 6e). All three genes displayed AS-induced 3′UTR shortening and the full 3′UTRs cloned all contained APA sites. Using dual luciferase assays, we found that after stress constructs containing these 3′UTRs produced ~40–50% higher luciferase activities than the control construct lacking these 3′UTRs (Fig. 6f). In the case of *Nmt1*, we also found that its full 3′UTR, which had the ability of APA, led to 1.2-fold higher luciferase activities than the short 3′UTR (cUTR) in stressed cells (Fig. 6g), indicating the impact of APA on protein expression in response to stress.

Together, reporter assays confirmed that stress enhances proximal PAS usage in the cell, leading to increased expression of short 3′UTR isoforms that have higher protein expression potentials than long 3′UTR isoforms.

**Table 1 Top ten Gene Ontology (GO) terms enriched for genes with significant 3′UTR shortening through APA regulation by stress**

| P-value | GO term |
|---------|---------|
| 6.5E-04 | Cell differentiation |
| 2.3E-03 | Cellular component assembly |
| 2.9E-03 | Signal transduction |
| 3.6E-03 | Cell proliferation |
| 6.1E-03 | Circulatory system process |
| 1.0E-02 | Cell death |
| 1.2E-02 | Anatomical structure formation involved in morphogenesis |
| 1.6E-02 | Homeostatic process |
| 1.7E-02 | Anatomical structure development |
| 2.0E-02 | Immune system process |

Genes were selected based on newly made RNAs in AS (1 h) and RC (4 h) cells, corresponding to the blue dots in Fig. 5b. Only Biological Process terms are shown

**Distinct stress-induced APA profiles in different cell contexts**. The GO analysis result indicated that genes with functions in cell proliferation and differentiation tended to have their 3′UTRs shortened by stress (Table 1). We next asked how stress impacts 3′UTR lengths in cells in different proliferation and differentiation states, which have different APA programs[26]. To this end, we analyzed APA by 3′READS in proliferating C2C12 myoblast (MB) cells and differentiated myotube (MT) cells under normal, stressed (250 μM NaAsO$_2$ for 1 h), and recovery (4 h after AS) conditions (Fig. 7a).

As in NIH3T3 cells, AS elicited significant 3′UTR shortening in both MB and MT cells to a similar extent after 1 h of stress, with median RED = −0.14 and −0.15 for MB and MT cells, respectively (Fig. 7b). Also similar to NIH3T3 cells, both MB and MT cells showed more significant 3′UTR shortening after 4 h of recovery, with MB cells showing a greater extent of 3′UTR shortening (median RED = −0.48) than MT cells (median RED = −0.31) (Fig. 7b). Note that these 3′UTR size changes are greater in scale than the 3′UTR size changes taking place in cell differentiation (median RED = −0.19, MB vs. MT, Supplementary Figs. 6a and 6b), highlighting the substantial impact of stress on 3′UTR landscape in the cell. A modest correlation in 3′UTR changes could be discerned between MT and MB cells during recovery from stress (r = 0.37, Pearson correlation, Supplementary Fig. 6c).

We next asked whether stress-regulated 3′UTR APA events were related to those modulated in cell proliferation/differentiation (P/D). To this end, we compared 3′UTR APA changes between MT and MB cells (representing P/D) with those between RC and NT samples of MT and MB cells. A mild correlation could be discerned between APA changes in MT vs. MB and those by stress (r = 0.13, Pearson correlation, Fig. 7c). We then divided genes into four groups based on 3′UTR shortening in the two conditions: both P/D and stress (Both), P/D only, Stress only, and neither P/D nor stress (Other) (Fig. 7c). Interestingly, genes in the Both group tended to have longer aUTRs (median = 1759 nt, Fig. 7c) as compared to genes in the Stress only group (median = 788) or in the P/D only group (median = 1103). By contrast, the median aUTR size was the shortest (397 nt) for genes in the Other group. This result indicates that long aUTR may confer regulability under both P/D and stress conditions.

We found that genes in the Both and Stress only groups showed significant 3′UTR shortening in stressed NIH3T3 cells as well, as compared to genes in the Other group (P = 7.8 × 10$^{-15}$ and = 5.3 × 10$^{-15}$, respectively, Fig. 7d). By contrast, no difference could be discerned between genes in the P/D only group and

those in the Other group (Fig. 7d), highlighting the distinction between stress and P/D in 3′UTR length regulation.

Using RT-qPCR, we validated several genes in different groups (Fig. 7e). Nmt1 showed 3′UTR regulation in both P/D and stress. Timp2 also showed 3′UTR regulation in P/D, but its regulation by stress was restricted to MB cells only. Rpl22 and Fam49b showed substantial 3′UTR shortening in P/D but modest shortening by stress, whereas 3′UTR shortening of Dnajb1 and Hspa4l only occurred in stressed cells. Together, these results indicate that stress-induced 3′UTR shortening takes place in both proliferating and differentiated cells, with proliferating cells showing a greater extent of changes. While 3′UTR length controls by P/D and stress are largely distinct, some genes have their 3′UTRs regulated in both conditions.

## Discussion

In this study, we report that cellular stress can globally change the 3′UTR landscape through two distinct mechanisms. First, stress has an immediate impact on PAS usage, generally activating proximal PAS usage. This mechanism leads to shortening of 3′UTRs for many genes, especially those with functions in cell differentiation and proliferation. Second, long 3′UTR isoforms are preferentially degraded during recovery from stress by enhanced 3′UTR-based mRNA decay, further shortening 3′UTRs in the cell. We show that TIA1 plays an important role in the second phase of 3′UTR shortening through binding to U-rich motifs in alternative 3′UTR sequences, leading to SG recruitment during stress and mRNA decay in recovery. Importantly, genes that are capable of 3′UTR shortening through APA upon stress can effectively evade mRNA decay after stress, leading to preservation of transcripts. We thus conclude 3′UTR shortening by APA is an adaptive stress response to preserve mRNAs, especially transcripts of proliferation and differentiation genes.

While we show multiple lines of evidence supporting regulation of PAS choice by stress, the underlying mechanism is not clear. Previous studies have shown that oxidative stress can alter chromatin structure and epigenetic features[48]. While these mechanisms have been implicated in PAS usage regulation[49,50], our reporter assays based on episomal plasmids indicate that the enhanced proximal PAS usage in stressed cells is more likely to be caused by trans factors. We found that AS-induced APA changes correlate with aUTR sizes, indicating that the underlying mechanism involves competition between APA sites. This was previously observed in cells with knockdown of C/P factors[41]. However, after 1 h of AS, we did not observe a global expression increase of C/P factor mRNAs. Given the short window of AS-induced 3′UTR shortening (1 h), it is more plausible that translational control and/or post-translational modifications of some key C/P factors lead to enhanced cleavage/polyadenylation at proximal PASs. It is worth noting that our data are not incongruous with previous findings indicating inhibition of 3′ end processing by stress[51], because we analyzed relative usage of APA sites. Greater inhibition on distal PASs relative to proximal PASs, for example, could lead to more usage of proximal PASs, which has been shown for APA regulation by CFI-25/68 (refs. [41,52]).

Our result showing that long 3′UTR isoforms are more likely to be associated with SGs than short isoforms is line with a recent finding indicating that transcript length is a key determinant for RNA enrichment in the SG core[46]. However, our study further revealed several novel aspects of mRNA association with SGs, especially concerning TIA1: First, TIA1 binding with mRNAs correlates well with 3′UTR size and frequency of U-rich motifs. Consequently, due to the prominent presence of U-rich motifs in aUTRs, long and short 3′UTR isoforms differ substantially in TIA1 binding, leading to distinct mRNA metabolisms in normal

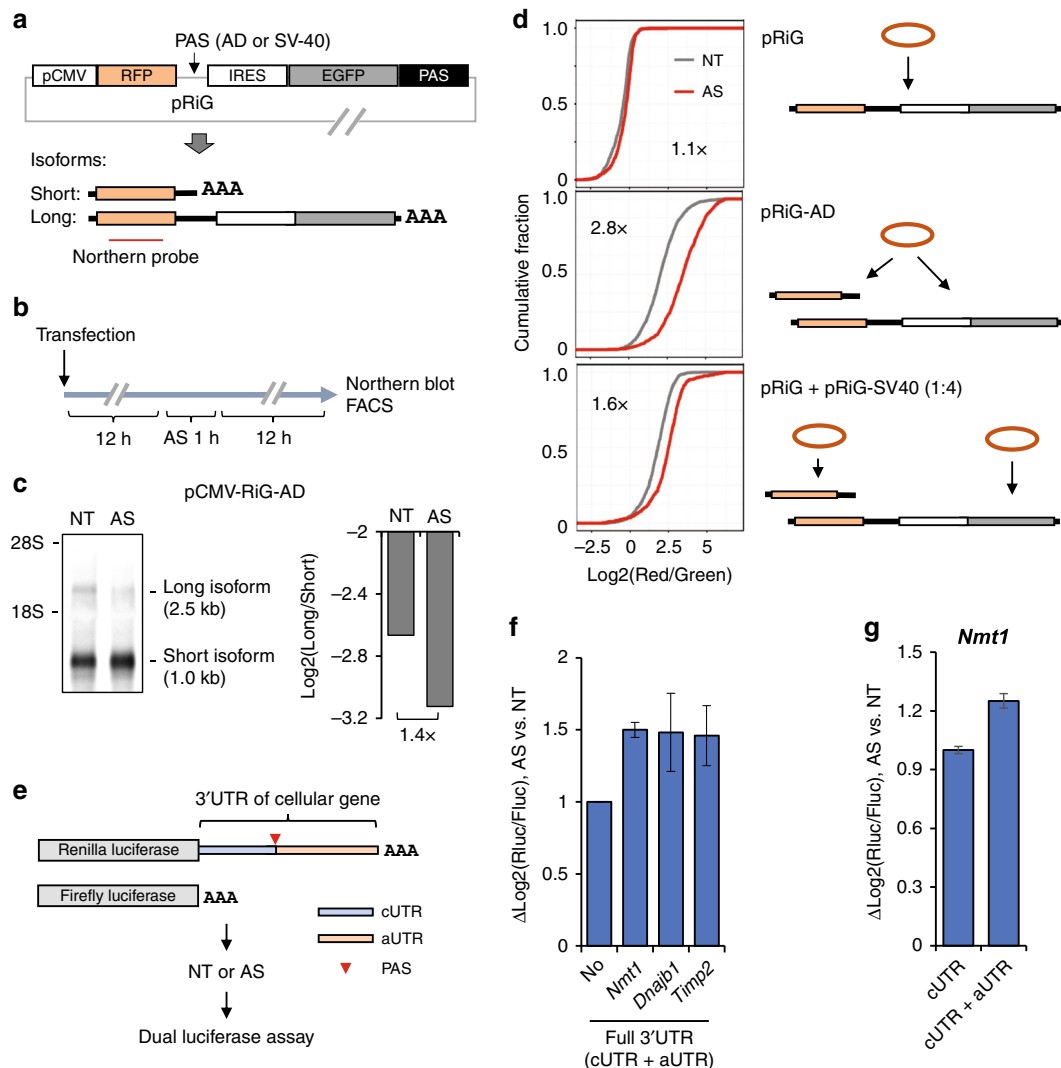

**Fig. 6** Reporter assays indicate that APA confers enhanced gene expression after stress. **a** Schematic of pRiG-based vectors for examining of PAS usage. pRiG-AD can generate two APA isoforms because of the presence of a weak PAS between RFP and IRES-EGFP regions, as indicated. pRiG does not contain a PAS and thus expresses the long isoform only. pRiG-SV-40 has a strong PAS, leading to expression of the short APA isoform only. RFP, red fluorescent protein; IRES, internal ribosome entry site; EGFP, enhance green fluorescent protein. Probe for Northern blot analysis is indicated. **b** Schematic showing experimental design. **c** Left, Northern blot analysis of two APA isoforms expressed from the pRiG-AD vector in NT or AS samples. Right, Log2(ratio) of expression level of long isoform to that of short isoform shown in the Northern blot. **d** Left, distribution of ratios of red fluorescence to green fluorescence in cells transfected with indicated plasmids in NT or AS cells. Higher log2(Red/Green) values indicate more expression of the short isoform encoding RFP. The difference in median log2(Red/Green) between NT and AS cells is indicated in each graph. pRiG + pRiG-SV40 is a mixture of the two plasmids with the indicated ratio in parenthesis. Right, schematic of expression of APA isoforms from indicated plasmids. **e** Schematic of psiCHECK2 reporters. Full 3′UTR sequences of mouse *Nmt1*, *Dnajb1*, or *Timp2* were inserted downstream of the Renilla luciferase CDS. Firefly luciferase activity from the same reporter was used for normalization. **f** Dual luciferase assay for analysis of the effects of inserted full 3′UTR sequences on expression of Renilla luciferase in AS vs. NT cells. Error bars are standard deviation based on three replicates. **g** Dual luciferase assay for analysis of the effect of inserted cUTR or full 3′UTR sequence of *Nmt1* on AS-induced regulation of expression of Renilla luciferase. Error bars are standard deviation based on two replicates

and stressed cells. By contrast, CDS length was found to be more important than 3′UTR length for G3BP1 binding[46]. Therefore, distinct pathways can lead to SG recruitment. In line with this notion, we found that a set of SG-enriched transcripts appear to be TIA1-independent, and some transcripts with strong TIA1 binding are not highly enriched in SGs (Fig. 4e). Second, our data indicate that the transcript repertoires TIA1 interacts with in normal and stressed cells are not globally different. In other words, the potential of a transcript to localize into SGs under stress is generally pre-determined before stress. Third, SG association can lead to mRNA decay during stress recovery in a

3′UTR-dependent manner. Degradation of SG-associated RNAs can be mediated through autophagy, where the whole SG is destroyed[53] and/or P-bodies, where RNAs are degraded through regular decay mechanisms[16]. However, long 3′UTR transcripts with U-rich motifs appear to be preferentially degraded after stress. Therefore, despite universal recruitment to SGs, transcripts can have quite different fates after stress. Further studies are needed to address mechanistic details of SG recruitment and mRNA decay pathways in the context of knockdown or over-expression of TIA1 or other SG-related RBPs[54]. In addition, given the difference in translational efficiency between short and long

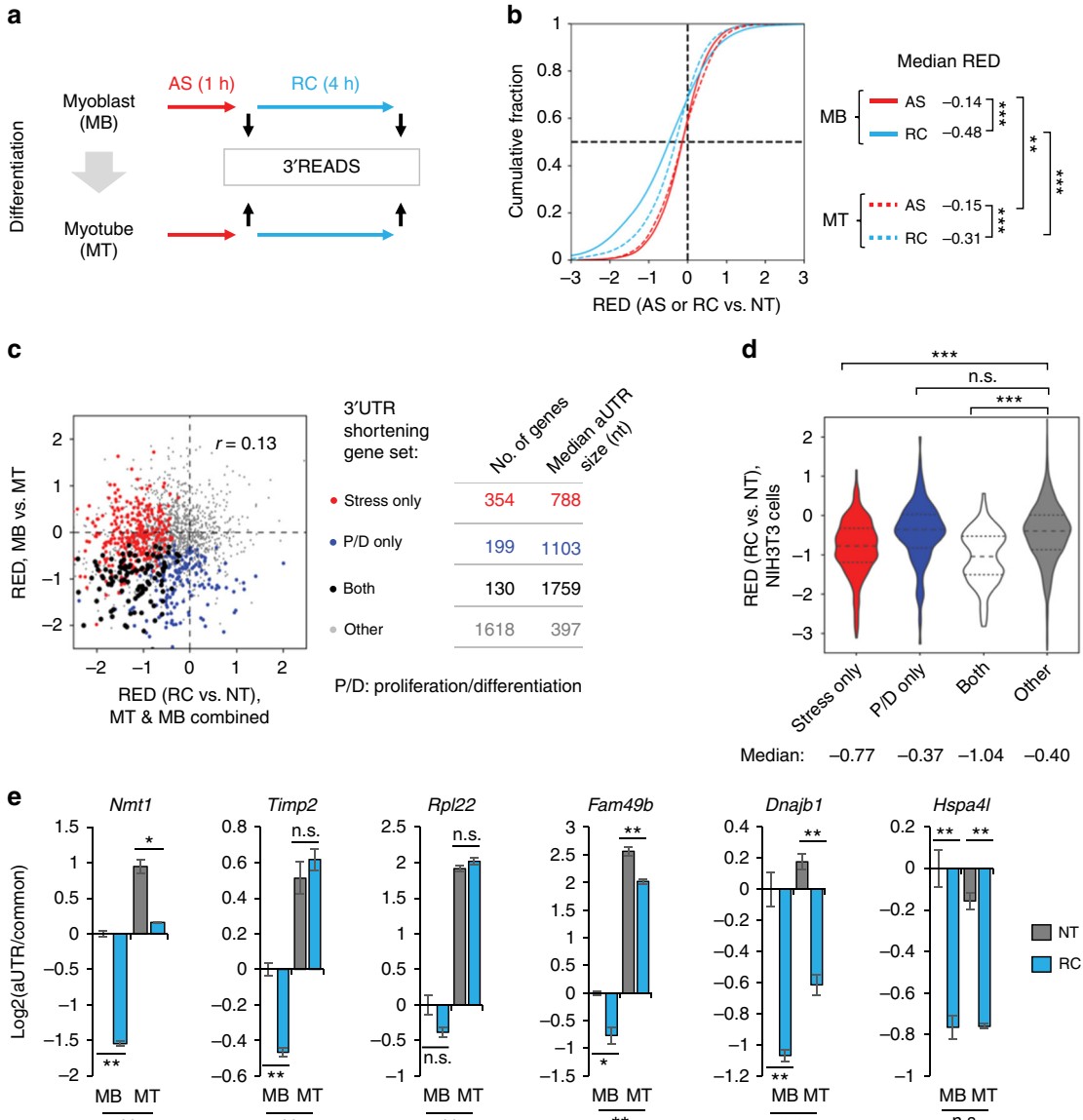

**Fig. 7** Stress-elicited 3′UTR shortening in proliferating and differentiated C2C12 cells. **a** Schematic of experimental design. **b** Distribution of 3′UTR APA changes (RED) in proliferating C2C12 myoblast (MB) or differentiated C2C12 myotube (MT) cells after 1 h of arsenic stress (AS) or 4 h of recovery after stress (RC) as compared to non-treated (NT) cells. Median RED for each comparison is shown, reflecting overall 3′UTR size change. The *P*-value (K–S test) indicating significance of RED difference is shown for each comparison. Three asterisks indicate *P* < 0.001 and two *P* < 0.01. **c** Comparison of stress-elicited 3′UTR APA changes in MT or MB cells (*x*-axis) and C2C12 proliferation/differentiation (P/D, *y*-axis). Pearson correlation coefficient is indicated. For *x*-axis, RED values of MT and MB cells were averaged. For P/D, two biological replicated were used. Genes were divided into four groups based on 3′UTR size changes in stress and P/D conditions, i.e. Both, Stress only, P/D only, and Other. Genes with significant 3′UTR shortening by stress were based on either MT or MB cells (a union set). Number of genes in each group and median aUTR size are shown. **d** Violin plots showing RED values in stressed NIH3T3 cells (4 h RC vs. NT, Fig. 1) for the four gene groups in **c**. The median of each is indicated. Significance of difference (Wilcoxon test) is indicated for comparing groups (***, *P* < 0.001; n.s., *P* > 0.05). **e** RT-qPCR analysis of 3′UTR size changes by stress in MT or MB cells. Error bars are standard deviation based on two replicates. The significance of difference between MT and MB cells in the NT condition is also indicated. One asterisk indicates *P* < 0.1, two asterisks *P* < 0.05, and n.s. *P* > 0.1 (Student's *t*-test)

3′UTR isoforms[43], it is also important to examine how 3′UTR shortening after stress may impact protein expression.

We show that the impact of stress on 3′UTR length varies between proliferating and differentiated cells. This might be due to different APA regulation and/or mRNA decay mechanisms in cells in different states. Because 3′UTR length regulation is part of gene expression program in cell proliferation, differentiation and development[25,26], stress-induced 3′UTR shortening may hamper

cell differentiation and development, which normally involve lengthening of 3′UTRs. In line with this, previous studies have shown that arsenic exposure delays cell differentiation[55,56] and affects development[57]. Similarly, shortening of 3′UTR has previously been associated with cancer development through activation of proto-oncogene expression[58]. Therefore, one open question is how stress-elicited 3′UTR shortening may play a role in stress-induced oncogenesis[1].

## Methods

**Cell culture**. NIH3T3 cells from ATCC were cultured in high glucose Dulbecco's modified Eagle's medium (DMEM) supplemented with 10% calf serum. C2C12 cells from ATCC were cultured in DMEM supplemented with 10% fetal bovine serum. C2C12 cell differentiation was induced by changing culture media to DMEM supplemented with 2% horse serum when cells were at >95% confluency. All media contained 100 IU/ml penicillin and 100 μg/ml streptomycin. All cells were incubated at 37 °C with 5% $CO_2$. To induce stress, cells were incubated in culture medium containing 250 μM or 25 μM of sodium arsenite ($NaASO_2$), or 1 mM of $H_2O_2$. For stress recovery, stressed cells were washed twice with phoshate-buffered saline (PBS) to remove $NaASO_2$ or $H_2O_2$, followed by culturing in regular media for a period of time before harvest. Cell viability assay and cell counting were performed using Vi-CELL Cell Viability Analyzer (Beckman Coulter).

**Plasmids**. pRiG, pRiG-SV40 and pRiG-AD were described in ref. [26]. psiCHECK2-3′UTR constructs were made by inserting PCR-amplified mouse 3′UTR sequences into the psiCHECK2 plasmid (Promega) linearized with XhoI and NotI. The PCR primer sequences are listed in Supplementary Table 3.

**Dual luciferase assay**. Twelve hours after transfection with reporter plasmids using Lipofectamine 3000 (Thermo Fisher), NIH3T3 cells were treated with 250 μM sodium arsenite for 1 h. Cells were then washed twice with PBS and cultured in regular medium for another 12 h. Cells lysate was prepared, and dual luciferase assay was performed using Dual-Luciferase Reporter Assay System (Promega) following the manufacturer's protocol. Biological triplicates were performed.

**Immunofluorescence staining and microscopy**. Cells grown on glass coverslips were fixed using 4% paraformaldehyde in PBS for 10 min at room temperature and permeabilized with PBS containing 0.1% Triton X-100 at room temperature for 10 min. After blocking with 1% bovine serum albumin in PBST (PBS + 0.1% Tween 20) for 30 min, SGs were immunostained with rabbit anti-PABPC1 (Abcam, ab21060, 1:500 in PBST) or goat anti-TIA1 (Santa Cruz Biotechnology, sc-1751, 1:500 in PBST) for 60 min. FITC-conjugated goat anti-rabbit antibody (Jackson ImmunoResearch, 111-095-144, 1:500 in PBST) or Texas red-conjugated donkey anti-goat (Jackson ImmunoResearch, 705-075-147, 1:500 in PBST) was applied at RT for 60 min. Cells were mounted on glass slides in SlowFade Gold Antifade reagent (Thermo Fisher) with DAPI. Fluorescence images were collected using the EVOS FL Auto Cell Imaging System (Thermo Fisher).

**Ribonucleoprotein immunoprecipitation**. The protocol for isolation of TIA1-associated RNA was adapted from the RIP protocol described in Keene et al.[59] Briefly, NIH3T3 cells with or without 1 h of AS were trypsinized, washed twice with pre-chilled PBS, re-suspended and incubated in lysis buffer (20 mM HEPES-KOH pH 7.5, 15 mM $MgCl_2$, 80 mM KCl, 1% Triton X-100, 2 mM DTT, 100 μg/ml cycloheximide, 1× SIGMAFAST protease inhibitor cocktail, and 200 U/ml Super-aseIn) on ice for 10 min. During the incubation, cells were gently sheared three times through a pre-chilled 26-gauge needle. The cell lysate was centrifuged at $700 \times g$ at 4 °C for 5 min to pellet nuclei. The supernatant was transferred to a new tube and centrifuged at $14,000 \times g$ at 4 °C for 5 min. About 10% of the supernatant was saved for cytoplasmic RNA extraction, while the rest of the supernatant was diluted five times with NT2 buffer (50 mM Tris, pH 7.4, 150 mM NaCl, 1 mM $MgCl_2$, 0.05% NP-40) supplemented with 100 U/ml SuperaseIn. The diluted lysate was incubated with goat anti-TIA1 IgG (Santa Cruz, sc-1751) conjugated to Dynabeads Protein G at 4 °C for 4 h. After washing the beads five times with NT2 buffer supplemented with 1 M urea, RNA was extracted from the beads and the input cytoplasmic fraction using Trizol, followed by 3′READS analysis. Alternatively, proteins were eluted from the beads following the manufacturer's protocol and analyzed by western blot analysis using rabbit anti-TIA1 (Proteintech, 12133-2-AP). Uncropped blots can be found in Supplementary Fig. 7d.

**3′READS library construction and sequencing**. The 3′READS procedure (3′READS+ version) was described in ref. [40]. Briefly, Poly(A)+ RNA in 0.1 or 1 μg of input RNA was captured using oligo(dT)$_{25}$ magnetic beads (NEB) and fragmented on the beads using RNase III (NEB). After washing away unbound RNA fragments, poly(A)+ RNA fragments were eluted from the beads and precipitated with ethanol, followed by ligation to heat-denatured 5′ adapter (5′-CCUUGGCACCCGAGAAUUCCANNN) with T4 RNA ligase 1 (NEB). The ligation products were captured by biotin-T$_{15}$-(+TT)$_5$ (Exiqon) bound to Dyna-beads MyOne Streptavidin C1 (Thermo Fisher). After washing, RNA fragments on the beads were incubated with RNase H to remove bulk of the poly(A) tail and then eluted from the beads. After precipitation with ethanol, RNA fragments were ligated to a 5′ adenylated 3′ adapter (5′-rApp/NNNGATC GTCGGACTGTAGAACTCTGAAC/3ddC (Bioo Scientific) with T4 RNA ligase 2 (truncated KQ version, NEB). The ligation products were then precipitated and reverse transcribed using M-MLV reverse transcriptase (Promega), followed by PCR amplification using Phusion high-fidelity DNA polymerase (NEB) and bar-coded PCR primers for 15–18 cycles. PCR products were size-selected twice

with AMPure XP beads (Beckman Coulter). The size and quantity of the libraries were examined on an Agilent Bioanalyzer. Libraries were sequenced on an Illumina NextSeq 500 (1 × 75 bases).

**Fractionation of newly made and pre-existing RNAs**. Cells were cultured in medium supplemented with 50 μM of 4-thiouridine (4sU; Sigma) for 1 h before harvest. Total RNA was extracted using TRIzol. Newly made (4sU-labeled) and pre-existing RNA populations were fractionated following the protocol described in ref. [42]. Briefly, 100 μg of total RNA was biotinylated using biotin-HPDP (1 μg/μl in DMF; Thermo Fisher Scientific), and then extracted with chloroform three times and precipitated with ethanol. The biotinylated RNA was captured by Streptavidin C1 Dynabeads. The beads were washed six times, and the unbound, flow-through (Ft) RNA was collected. Bound, biotinylated RNA was eluted by DTT. All RNA was precipitated with ethanol and then used for RT-qPCR or 3′READS experiments.

**Strand-specific RNA-seq**. Poly(A)+ RNA in 5 μg of total RNA was purified twice using oligo(dT)$_{25}$ magnetic beads (NEB) following the manufacturer's protocol. RNA was then fragmented by partial alkaline hydrolysis (94 °C for 2 min), 3′ dephosphorylated with shrimp alkaline phosphatase (NEB), and 5′ phosphorylated with T4 PNK (NEB). The RNA fragments were sequentially ligated to a 5′ ade-nylated 3′ blocked 3′ adapter (5′-Aden/NNNNNNGATCGTCGGACTGT AGAACTCTGAAC/3ddC, where N is a random nucleotide) with truncated RNA ligase 2 (KQ, 200 U/μl) and to a 5′ adaptor (5′-CCUUGGCACCCGAGAA UUCCA) with T4 RNA ligase I. After reverse transcription of ligation products, cDNA amplification by PCR, and library purification by AMPure beads (Beckman Coulter), libraries were sequenced on an Illumina NextSeq 500 (1 × 75 bases).

**Fluorescence-activated cell sorting**. Twelve hours after transfection with reporter plasmids using Lipofectamine 3000, NIH3T3 cells were treated with 250 μM sodium arsenite for 1 h. Cells were then washed twice with PBS and cultured in regular medium for another 12 h. Cells were released from culture dishes by Trypsin-EDTA. Green and red fluorescent signals were read at 530 and 585 nm, respectively, in a BD FACScalibur system (BD Biosciences). Un-transfected cells were used to determine background fluorescence.

**Northern blot analysis**. Northern blot analysis was performed using the NorthernMax$^{TM}$ kit reagents (Thermo Fisher). Briefly, 12 h after plasmid trans-fection using Lipofectamine 3000, NIH3T3 cells were treated with or without 250 μM sodium arsenite for 1 h. After washing with PBS twice, cells were cultured in regular medium for another 12 h. Total RNA was extracted using TRIzol reagent (Thermo Fisher). A total of 5 μg of RNA was loaded onto a 1% formaldehyde-agarose gel for electrophoresis. RNA was then transferred onto a nylon membrane (Thermo Fisher). For hybridization, DIG-labeled DNA probes were generated by PCR using RFP-specific primers and the PCR DIG Probe Synthesis Kit (Sigma). The DIG-labeled probes were used for overnight hybridi-zation with the NorthernMax$^{TM}$ Kit. Immunological detection of the DIG-labeled probes was performed using the DIG Northern Starter Kit (Sigma). Signals on the blot were detected by a ChemiDoc Touch Imaging System (Bio-Rad). Images were quantified using the Fiji program[60]. Uncropped blots can be found in Supplementary Fig. 7b.

**Real-time PCR analysis**. cDNA was synthesized using the M-MLV reverse tran-scriptase (Promega), oligo(dT) primer, and 2 μg of RNA that was pre-treated with Turbo DNase (Thermo Fisher). cDNA was then mixed with primers and Hot Start Taq-based Luna qPCR master mix (NEB). PCR was run on an StepOne Plus Real Time PCR system (Thermo Fisher). For analysis of APA isoforms, primers were designed to amplify a region common to both pPAS and dPAS isoforms (com-mon), and a region between the two PASs (aUTR). Relative expression of dPAS isoform to pPAS isoform was calculated by log2(aUTR/common). Two-tailed t-test (without assuming equal variance) was performed to calculate P-values. qPCR primers are listed in Supplementary Table 1.

**Immunoblotting**. Protein concentration was measured using the DC Protein Assay (Bio-Rad). A total of 20 μg of protein per sample was resolved by sodium dodecyl sulfate polyacrylamide gel electrophoresis, followed by transfer to PVDF mem-brane for immunoblotting. Signals on the blot after adding the ECL reagent (Bio-Rad) were detected by the ChemiDoc Touch Imaging System (Bio-Rad). Quanti-fication was carried out in using the Fiji program[60]. Pre-stained protein molecular weight marker (Thermo Scientific, #26612) was used for protein size calculation. Uncropped blots can be found in Supplementary Figs. 7a, 7c and 7d.

**Analysis of 3′READS data**. 3′READS data were analyzed using methods described previously[40]. Briefly, the 5′ adapter sequence was first removed using Cutadapt[61]. Reads with short inserts (<23 nt) were discarded. The retained reads were mapped to the mouse genome (mm9) using bowtie2 (local mode)[62]. The three or six random nucleotides at the 5′ end (from the 3′ adapter) were removed before

mapping using the setting "−5 3" or "−5 6" in bowtie2. Reads with a mapping quality score (MAPQ) ≥10 were kept for further analysis. Reads with ≥2 non-genomic 5′Ts after alignment were called PAS reads. PASs within 24 nt from each other were clustered. PAS read counts mapped to genes were normalized by the median ratio method in DESeq[63]. Only APA isoforms with read count greater than five in the pair of compared samples were used. The two most abundant APA isoforms (based on PAS reads) in the 3′UTR of the 3′-most exon were selected for 3′UTR APA analysis. Significant APA events were those with a relative abundance change >5% and $P$-value <0.05 (Fisher's exact test). RED was calculated as the difference in log2(ratio) of abundances of two PAS isoforms (dPAS vs. pPAS) between two samples. To statistically assess the significance of overall APA differences between samples with different sequencing depths, we used the GAAP method we previously developed[41]. Briefly, one million PAS reads from each sample were randomly sampled through bootstrapping, and a median RED was calculated each time using all sampled genes. Sampling was carried out 20 times, and the mean of median RED and standard deviation were calculated.

**Analysis of RNA stability**. For each transcript with a defined PAS, its expression value (RPM) in the flow-through (Ft) sample was divided by the RPM value in the 4sU sample. The resulting value, log2(Ft/4sU), was adjusted to control for the number of uracils (Us) in each transcript because of its effect on labeling and capture efficiencies of 4sU-labeled RNA. A linear regression model for log2 (number of Us) vs. log2(Ft/4sU) was constructed by a python function. The adjusted log2(Ft/4sU) was calculated using the model.

**Analysis of TIA1 binding**. TIA1 binding efficiency for each PAS-defined transcript was calculated by log2 (RPM in the TIA1 RIP sample/RPM in the input sample). The averaged binding efficiency of each transcript based on two replicates was used for analysis. Linear regression models using transcript features, including 5′UTR size, CDS size, 3′UTR size, number of exons and exon density, againt log2(TIA1/input) were constructed with the python program 'scikit-learn'.

**Analysis of SG-enrichment data**. SG-enrichment information of human transcripts was obtained from the supplementary file in ref. [46]. TIA1 binding efficiency of the most abundant transcript of each mouse gene was compared to the SG-enrichment score of its human orthologue based on NCBI HomoloGene database[64].

**Analysis of RNA-seq data**. Raw reads from RNA-seq were first trimmed for adapter sequences by Cutadapt[61] and then mapped to the mm9 genome assembly using STAR (v2.5.2)[65] with default parameters. The read count of the coding sequence (CDS) of each gene was summed and then normalized by the median ratio method in DESeq[63]. Only genes with more than five reads in all samples were used.

**Gene Ontology analysis**. The GO terms and their associations with mouse genes were obtained from the Gene Ontology database[66]. The Fisher's exact test was used to derive $P$-values to indicate significance of association between a gene set and a GO term.

**Analysis of TIA1 iCLIP data**. Genomic positions and corresponding read counts of TIA1 iCLIP sites were obtained from GSE93575 (ref. [44]). iCLIP sites were further clustered within a 6-nt or 30-nt window. iCLIP sites with fewer than 10 reads were not used in analysis.

Assignment of TIA1 binding sites to cUTRs and aUTRs was based on mouse mammal-conserved PASs in PolyA_DB 3 (ref. [45]).

**Motif analysis of aUTR sequences**. Tetramer frequencies in aUTRs were calculated and compared between gene sets. $P$-values based on the Fisher's exact test were used to indicate the significance of enrichment.

**Code availability**. All scripts used for data processing and statistical analysis were written in Python, Perl, or R, and are available upon request.

**Data availability**. Sequencing datasets generated in this study, including those by 3′READS and RNA-seq (listed in Supplementary Table 2), have been deposited into the GEO database under the accession number GSE101851 [https://www.ncbi.nlm.nih.gov/geo/query/acc.cgi?acc=GSE101851]. All other data supporting the findings in this study are available from the corresponding author on reasonable request.

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

## Acknowledgements

We thank members of BT lab for helpful discussions. This work was funded by the NIH grant GM084089 to B.T.

## Author contributions

D.Z. and B.T. conceived of and designed the experiments. D.Z., T.W., B.X., Q.D. and L.W. performed the experiments. R.W. and D.Z. analyzed the data. Z.Z. contributed to reagents and materials. D.Z., R.W. and B.T. wrote the paper.
