## [Peer Review File · Nature Communications]

Reviewers' comments:

Reviewer #1 (Remarks to the Author):

This paper by Zheng et al. describes the finding that arsenic stress causes global changes in 3' UTR length, and thereby affects the expression of a particular set of genes during stress and stress recovery.

A number of important findings are reported. 3' READS uncovers a significant shortening of 3' UTRs of hundreds of genes upon arsenic stress, and, surprisingly, this shortening is enhanced further during recovery from stress. Distinct processing mechanisms are used to account for shortening in stress and recovery: while APA is the main mediator of shortening during stress, a differential half-life of long forms mostly accounts for the higher abundance of shorter 3'UTRs during recovery. Both APA and half-life are affected most in genes that display a longer aUTR (larger difference dPA/pPA). RIP-READS on the protein Tia1 shows that this protein binds preferably to long UTR isoforms, and Tia1 binding in cells in recovery correlates with long UTRs and a low half-life. The authors show evidence for a stabilisation of short isoforms during recovery of those genes that undergo shortening; the authors propose APA as a mechanism to evade mRNA decay in stress conditions.

Interestingly, genes with AS-induced shortening are often involved in cell proliferation/differentiation, and the authors go on to comparing the effect of cellular state (proliferation vs. differentiation) and AS. They find that the genes showing the most shortening in stress and those showing the most shortening in P/D carry particularly long UTRs. It is of note that stress has a greater, perhaps more immediate, effect on 3'UTR shortening than cellular state (proliferation/differentiation). However some features of the regulation by these two conditions are common, such as the preference for long 3' UTRs.

While this paper shows some interesting data and novel insight on APA as well as RNA stability in stress and proliferation vs. differentiation, I feel that key points are underdeveloped or insufficiently supported by experimental data.

One major claim is that the increased 3'UTR shortening during recovery (vs. AS) may be due to destabilisation of long RNAs, which does not take place during stress. However, many mRNAs are known to be stabilised during stress. Hence, the described destabilisation during recovery may actually reflect the reversal of stabilisation during stress. The authors should refer to other studies and discuss their findings in that context.

The authors show that SG enrichment from published datasets correlates with Tia1 binding; this is insufficient evidence to show involvement of Tia1 in SG recruitment. If 3'UTR length is a determinant for SG recruitment (as shown by others), and 3'UTR length is a determinant for Tia1 binding (as shown here), the correlation is hardly surprising. The authors should show that Tia1 indeed recruits long 3' UTRs to stress granules. For example,

- show that Tia1 localizes to SG, and
- show that knockdown of Tia1 reduces localization of a long UTR isoform, but not a short one (reporter gene or endogenous) within SG.

In the reporter assay, IRES and GFP sequences mimic a long 3'UTR, hence there are no bona fide 3'UTR sequences. While the experiments support the claim that stress can elicit APA, it does not substantiate the claim that cellular mRNAs with short 3' UTRs have greater protein expression potential. I suggest the authors support their conclusions using a reporter carrying bona fide 3' UTRs, for example of a few genes that they found most regulated in stress conditions. In such an assay, it would also be possible to test RNA stability vs. APA during stress recovery, and possibly a role of Tia1 in RNA stabilisation of shortened UTRs during recovery (or destabilisation of long UTRs).

While this study mainly focuses on genes that undergo 3' UTR shortening, there is a substantial set of genes that follow the opposite trend (roughly 30% of all genes undergoing AS-induced APA). It would be of interest to analyse these 3'UTRs, possibly to find whether some regulatory elements enriched in shortened 3'UTRs are depleted in aUTRs (and vice versa), as well as to analyse their GO.

Minor comments

Introduction:

- Fix the syntax of the sentence: ... tend to have the longest 3'UTRs than other tissue types

Results:

- Mention in the text or legend of Fig. 1 that you wash the arsenite is washed out for stress recovery
- Total eIF-2a levels seem to increase during stress recovery (Fig. 1B). Please comment
- Fig. 1F: indicate that the red line is stress and green recovery in the figure.
- Switching Figures 2B and 2C, and modify the text accordingly, would improve clarity. 2C: shortening of newly made RNAs in stress and recovery, 2B: comparison with existing RNAs and interpretation on RNA stability. Fig. 2B already indicates that long, existing RNAs are destabilized in recovery, while in AS, those long forms are stabilized.
- Fig. 3: show western blot of IP efficiency and specificity

Reviewer #2 (Remarks to the Author):

In this manuscript, the authors work on alternative polyadenylation(APA) in stressed cells and recovery from stress. They reported that arsenic stress can induce globally shorten UTR3, and proposed two mechanisms for stress induced 3'UTR shortening. While the stress induced alternative splicing of mRNA was previously reported, this work is probably the first thorough study of how cellular stress affects global regulation, which is another main mechanism for alterative RNA procession. Mechanistically, they found that the RNA-binding protein Tia1 can interact with long 3'UTR isoforms via U-rich elements and help the formation of stress granule. The findings are pretty novel (although not surprising) and the data are useful in understating RNA processing during cellular stress.

Major points:

Page 4, it is unclear what aUTR means, is it average UTR? The author should give a discussion on why the shortening is correlated with UTR size? Is this simply because longer UTR have more "room" to shorten, which means the relative shortening will be more or less constant?

Stress will usually induce translation inhibition for most genes, as shown by the phosphorylation of eIF2 α . ON the same time, mRNA with shorter 3'UTR usually have stronger activity in driving translation. The author should discuss or speculate if the global shortening of 3'UTR may have any effect on mRNA translation, if the mRNA "protected" by 3UTR shortening are those with high translation tendency (i.e. polysome occupancy).

Minor point:

There seems to be a mistake in Figure 2B where the Y-axis was labeled wrong.

Reviewer #3 (Remarks to the Author):

The authors of this paper performed a thorough and potentially very interesting analysis of the effects of arsenite induced stress on alternative polyadenylation. They have started off with the NIH3T3 model and show that arsenite treatment induces shortening of 3'UTRs in particular in freshly synthesized mRNAs in cells recovering from stress. The authors show bioinformatic data indicating that TIA binding to long 3'UTRs and recruitment to stress granules and degradation of such mRNAs may be the mechanism underlying this preferential mRNA destabilization. This hypothesis is tested by an analysis of reporter construct mRNA and protein expression. Further, the authors posit on the basis of GO-term analysis of mRNAs with short 3'UTRs that are stable during the recovery phase that genes governing proliferation and differentiation are preferential targets of this mechanism, which is validated in precursor and differentiated muscle cell models (although these analyses appeared to have resulted in different gene sets than those that are affected by arsenic induced stress).

The interesting model proposed here thus posits that different lengths of the 3'UTR with differential TIA binding determine the fate of the mRNAs following stress and the contribution of preferentially stabilized mRNAs with short 3'UTRs during the recovery phase of stress. In order to substantiate the claims made in this manuscripts I recommend addressing the following points.

(1) The concentration of 250 μ M NaAsO₂ is unusually high and pleiotropic effects must be excluded by (a) performing key experiments such as those shown in Fig 1 at the lowest possible concentration that is shown to induce eIF2 α phosphorylation and (b) by controlling for cell viability/induction of apoptosis for example by measuring CASP activation at this low and at the high concentration used for the detailed further analyses.

(2) The specificity of the GO term analysis for mRNAs with shortened 3'UTRs shown in table 1 should be complemented by the same analysis for unchanged and and lengthened mRNAs.

(3) The findings with arsenite induced stress should be generalized by using other stressors such as heat or DTT.

(4) As the reporter experiments are used to validate the hypothesis of TIA-induced destabilization of mRNAs with long 3'UTRs, the abundance of TIA binding sites and TIA binding should be assessed in the 3'UTRs of the reporter mRNAs

Response to reviewers' comments

First and foremost, my colleagues and I would like to thank all the reviewers for their time and effort in reviewing our manuscript. We appreciate their positive and insightful comments. We have carried out additional experiments and analyses according to their suggestions. We believe the paper is now much strengthened. We hope our revision is to their satisfaction. Below are our point by point response to their comments.

###

Reviewer #1 (Remarks to the Author):

One major claim is that the increased 3'UTR shortening during recovery (vs. AS) may be due to destabilisation of long RNAs, which does not take place during stress. However, many mRNAs are known to be stabilised during stress. Hence, the described destabilisation during recovery may actually reflect the reversal of stabilisation during stress. The authors should refer to other studies and discuss their findings in that context.

We thank the reviewer for this insightful suggestion. We now discuss this in Discussion. We did in fact observe stabilization of mRNAs during stress, as we indicated in the original manuscript (Figure 2d). We also suggested in our original manuscript that shortening of 3'UTR during recovery is likely due to stress granule clearance. This implies a connection between destabilization during recovery and stabilization during stress, likely through stress granules. However, our new data show that destabilization is specific for U-rich elements (Figure 2f), indicating distinct pathways for SG association and decay. We have now made this point more clear in the revised manuscript.

The authors show that SG enrichment from published datasets correlates with Tia1 binding; this is insufficient evidence to show involvement of Tia1 in SG recruitment. If 3'UTR length is a determinant for SG recruitment (as shown by others), and 3'UTR length is a determinant for Tia1 binding (as shown here), the correlation is hardly surprising. The authors should show that Tia1 indeed recruits long 3' UTRs to stress granules. For example,

- show that Tia1 localizes to SG, and
- show that knockdown of Tia1 reduces localization of a long UTR isoform, but not a short one (reporter gene or endogenous) within SG.

The reviewer raised an excellent point. TIA1's role in recruitment of mRNAs into SGs is well documented, for example in Gilks et al. MBC, 2004. As suggested by the reviewer, we now show TIA1 enrichment in SGs during stress in our study using immunofluorescence (Figure S4a). In addition, we show that preferential binding of TIA1 to long 3'UTR isoforms is attributable to U-rich motifs in alternative 3'UTR sequences (Figure 3e). We further show that, based on a recently published iCLIP data set, about 80% of Tia1 binding sites in 3'UTRs are located in alternative UTR regions, which are present in long 3'UTR isoforms but not in short 3'UTR isoforms (Figure 3g). Moreover, we show differences between TIA1 binding and G3BP1 binding (Figure 4e). Therefore, there appears to be distinct pathways to recruit transcripts into SGs. We appreciate the suggestion to knock down TIA1 and examine long UTR isoform localization. But this is a tricky experiment due to the presence of TIA1 paralog TIAR which has similar roles to TIA1. We would like to pursue this in the future, with double knockdown or overexpression experiments.

Gilks, N., Kedersha, N., Ayodele, M., Shen, L., Stoecklin, G., Dember, L.M., and Anderson, P. (2004). Stress granule assembly is mediated by prion-like aggregation of TIA-1. *Mol Biol Cell* 15, 5383-5398.

In the reporter assay, IRES and GFP sequences mimic a long 3'UTR, hence there are no bona fide 3'UTR sequences. While the experiments support the claim that stress can elicit APA, it does not substantiate the claim that cellular mRNAs with short 3' UTRs have greater protein expression potential. I suggest the authors support their conclusions using a reporter carrying bona fide 3' UTRs, for example of a few genes that they found most regulated in stress conditions. In such an assay, it would also be possible to test RNA stability vs. APA during stress recovery, and possibly a role of Tia1 in RNA stabilisation of shortened UTRs during recovery (or destabilisation of long UTRs).

Thanks for the suggestions. We have cloned *bona fide* 3'UTR sequences of three genes (*Nmt1*, *Dnajb1*, and *Timp2*) into the psiCHECK2 vector and carried out reporter assays (Figure 6e-f). All genes showed significant 3'UTR shortening during stress. The result is consistent with the pRiG reporter assays. In all cases, the construct with a full 3'UTR containing both cUTR and aUTR (thus having the ability of APA) gave rise to higher luciferase activities as compared to a control construct. In the case of *Nmt1*, we further compared full 3'UTR and cUTR. The result is also consistent. We appreciate the suggestion concerning additional studies of TIA1 on RNA stability. While this is important for elucidating detailed mechanisms, it will involve significant amount of time and resources. Our current version of the paper is already quite complete. We will plan to address this in the future.

While this study mainly focuses on genes that undergo 3' UTR shortening, there is a substantial set of genes that follow the opposite trend (roughly 30% of all genes undergoing AS-induced APA). It would be of interest to analyse these 3'UTRs, possibly to find whether some regulatory elements enriched in shortened 3'UTRs are depleted in aUTRs (and vice versa), as well as to analyse their GO.

We have done GO analysis but did not find significant GO terms associated with genes showing the opposite trend. This is indicated in the paper. Thanks for the suggestion.

Minor comments

Introduction:

- Fix the syntax of the sentence: ... tend to have the longest 3'UTRs than other tissue types

Fixed.

Results:

- Mention in the text or legend of Fig. 1 that you wash the arsenite is washed out for stress recovery

It is now indicated in the main text and mentioned in the Methods.

- Total eIF-2 α levels seem to increase during stress recovery (Fig. 1B). Please comment

The difference observed on Fig. 1b is due to sample loading difference. After normalization to tubulin, we only see very mild eIF2- α protein level changes (see below, based on two biological replicates). We thus do not think eIF2- α level is regulated by arsenic stress.

- Fig. 1F: indicate that the red line is stress and green recovery in the figure.

Fixed. Thanks for the suggestion.

- Switching Figures 2B and 2C, and modify the text accordingly, would improve clarity. 2C: shortening of newly made RNAs in stress and recovery, 2B: comparison with existing RNAs and interpretation on RNA stability. Fig. 2B already indicates that long, existing RNAs are destabilized in recovery, while in AS, those long forms are stabilized.

Done. Thanks for the suggestion.

- Fig. 3: show western blot of IP efficiency and specificity

We now include the Western Blot result for our antibody (see Figure S4b). As shown, the antibody is quite specific. We have also included immunofluorescence data (see Figure S4a) which indicates TIA1 enrichment in SGs. In addition, we have compared our 3'READS+RIP data with a recently published TIA1 iCLIP data set (Figure S4d). The results show a general consistency between the two methods despite that two different cell types were used.

###

Reviewer #2 (Remarks to the Author):

Major points:

Page 4, it is unclear what aUTR means, is it average UTR? The author should give a discussion on why the shortening is correlated with UTR size? Is this simply because longer UTR have more "room" to shorten, which means the relative shortening will be more or less constant?

We apologize for the unclear writing. We now explain what aUTR (alternative UTR) stands for in the text and figure. Because of competition between two alternative PASs and the first-come-first-served mechanism, shortening is correlated with the distance between PASs, as we previously reported (Li et al. 2005, PLoS Genet. 11:e1005166). We now mention this in Discussion.

Stress will usually induce translation inhibition for most genes, as shown by the phosphorylation of eIF2 α . ON the same time, mRNA with shorter 3'UTR usually have stronger activity in driving translation. The author should discuss or speculate if the global shortening of 3'UTR may have any effect on mRNA translation, if the mRNA "protected" by 3UTR shortening are those with high translation tendency (i.e. polysome occupancy).

We now discuss potential consequences in Discussion. Thanks for the suggestion.

Minor point:

There seems to be a mistake in Figure 2B where the Y-axis was labeled wrong.

Fixed.

###

Reviewer #3 (Remarks to the Author):

(1) The concentration of 250 μ M NaAsO₂ is unusually high and pleiotropic effects must be excluded by (a) performing key experiments such as those shown in Fig 1 at the lowest possible concentration that is shown to induce eIF2 α phosphorylation and (b) by controlling for cell viability/induction of apoptosis for example by measuring CASP activation at this low and at the high concentration used for the detailed further analyses.

We thank this reviewer for this excellent suggestion. We have examined cell viability and cell growth (Figure S1). While the cells after 1 h of treatment with 250 μ M NaAsO₂ showed much hindered cell proliferation (Figure S1b), they did not go into cell death (Figure S1a). We also found that a lower level of AS (25 μ M of NaAsO₂), which does not lead to eIF-2 α phosphorylation, also caused 3'UTR shortening (Figure S2). In addition, oxidative stress inducer H₂O₂, which did lead to eIF-2 α phosphorylation, also caused 3'UTR shortening (Figure S2).

(2) The specificity of the GO term analysis for mRNAs with shortened 3'UTRs shown in table 1 should be complemented by the same analysis for unchanged and lengthened mRNAs.

We have now done this analysis. No significant GO terms could be found for genes with lengthened 3'UTRs. This is now mentioned in Discussion.

(3) The findings with arsenite induced stress should be generalized by using other stressors such as heat or DTT.

We have analyzed stress by H₂O₂. The result is similar. More stress conditions will be examined in the future.

(4) As the reporter experiments are used to validate the hypothesis of TIA-induced destabilization of mRNAs with long 3'UTRs, the abundance of TIA binding sites and TIA binding should be assessed in the 3'UTRs of the reporter mRNAs .

We have carried out more rigorous analyses of TIA1 binding, including comparison of our 3'READS+RIP data with a recently published iCLIP data set (Figures 3f, 3g, and S4d), and motif analysis based on comparison of TIA1 binding with SG enrichment (Figure 4e). The aspect on TIA1 binding is much strengthened.

###

Sincerely,

Bin Tian, Ph.D.

Rutgers New Jersey Medical School

REVIEWERS' COMMENTS:

Reviewer #1 (Remarks to the Author):

The manuscript "Cellular stress alters 3'UTR landscape through alternative polyadenylation and isoform-specific degradation" by Bin Tian and colleagues was extensively revised through additions in the text, clarifications on points raised, and additional data. The manuscript has substantially improved. My comments were satisfactorily addressed, with one exception: Figure 3, to show specificity of the TIA-1 immunoprecipitation by Western Blot, i.e., showing input vs. IP and supernatant of the pull-down. I would recommend this important experiment be shown instead of a simple Western Blot of TIA-1 in different cellular conditions. In my opinion, the paper is ready for publication.

Reviewer #3 (Remarks to the Author):

The authors have now addressed the points that I have raised in my review of the first version of this manuscript

Point-by-point response to reviewers' comments

The manuscript "Cellular stress alters 3'UTR landscape through alternative polyadenylation and isoform-specific degradation" by Bin Tian and colleagues was extensively revised through additions in the text, clarifications on points raised, and additional data. The manuscript has substantially improved. My comments were satisfactorily addressed, with one exception: Figure 3, to show specificity of the TIA-1 immunoprecipitation by Western Blot, i.e., showing input vs. IP and supernatant of the pull-down. I would recommend this important experiment be shown instead of a simple Western Blot of TIA-1 in different cellular conditions. In my opinion, the paper is ready for publication.

We have now included the western blot for IP'd samples (Supplementary Figure 4b). As shown, the IP was highly successful.